# Workforce characteristics and interventions associated with high-quality care and support to older people with cancer: a systematic review

Jackie Bridges,[1,2] Grace Lucas,[1] Theresa Wiseman,[1,3] Peter Griffiths[1,2]

► Prepublication history and additional material are available. To view these files please visit the journal online (http://dx.doi.org/10.1136/bmjopen-2017-016127).

[1]Faculty of Health Sciences, University of Southampton, Southampton, UK
[2]NIHR CLAHRC Wessex
[3]The Royal Marsden NHS Foundation Trust

**Correspondence to**
Prof. Jackie Bridges;
jackie.bridges@soton.ac.uk

## ABSTRACT

**Objectives** To provide an overview of the evidence base on the effectiveness of workforce interventions for improving the outcomes for older people with cancer, as well as analysing key features of the workforce associated with those improvements.

**Design** Systematic review.

**Methods** Relevant databases were searched for primary research, published in English, reporting on older people and cancer and the outcomes of interventions to improve workforce knowledge, attitudes or skills; involving a change in workforce composition and/or skill mix; and/or requiring significant workforce reconfiguration or new roles. Studies were also sought on associations between the composition and characteristics of the cancer care workforce and older people's outcomes. A narrative synthesis was conducted and supported by tabulation of key study data.

**Results** Studies (n=24) included 4555 patients aged 60+ from targeted cancer screening to end of life care. Interventions were diverse and two-thirds of the studies were assessed as low quality. Only two studies directly targeted workforce knowledge and skills and only two studies addressed the nature of workforce features related to improved outcomes. Interventions focused on discrete groups of older people with specific needs offering guidance or psychological support were more effective than those broadly targeting survival outcomes. Advanced Practice Nursing roles, voluntary support roles and the involvement of geriatric teams provided some evidence of effectiveness.

**Conclusions** An array of workforce interventions focus on improving outcomes for older people with cancer but these are diverse and thinly spread across the cancer journey. Higher quality and larger scale research that focuses on workforce features is now needed to guide developments in this field, and review findings indicate that interventions targeted at specific subgroups of older people with complex needs, and that involve input from advanced practice nurses, geriatric teams and trained volunteers appear most promising.

## BACKGROUND

More than 60% of new cancers and more than 70% of cancer deaths occur in people over the age of 65 years in Europe and the USA.[1] Treatment outcomes for older patients with cancer vary internationally[2] and this may be linked to the extent to which services and their associated workforce effectively meet the more complex needs associated with an ageing population.[3 4] Many older people have comorbidities and limitations which affect their cognitive and physical functioning, their risk of complications and their emotional well-being,[5] all of which may affect cancer treatment tolerance and necessitate a modified treatment plan and relevant supportive care.[6] More comprehensive assessment and management has been recommended to optimise older patients with cancer for treatment.[6–8] Furthermore, older people may value a range of outcomes beyond survival at any cost, including maintaining independence and being able to access information, emotional support and practical support both during and after treatment.[9] Healthcare workers who organise and deliver cancer care thus need knowledge of clinical and other issues which are common in old age, but also need to be adept with the skills and values to enable them to support the patient and

family, develop treatment plans, deliver appropriate care and help older people to achieve the quality of life (QOL) that reflects what matters most to them as individuals.[10]

While the specific role of the healthcare workforce in ensuring optimal outcomes and QOL for older cancer survivors and their families has been recognised, evidence suggests that there are variations internationally in the preparedness of the workforce to meet the needs of an ageing population.[9–16] Issues identified include deficits in the necessary education, knowledge, skills and attitudes; in staffing levels and skill mix; and in the development of roles, teams and services that meet older people's needs.[17] However, little is known about the features and characteristics of the workforce associated with better outcomes for older people with cancer, or about the relative effectiveness of workforce-focused interventions which are aimed at improving cancer care and outcomes for an ageing population. This systematic review therefore aims to inform developments in policy and practice by providing an overview of the evidence base on the effectiveness of workforce interventions for improving the outcomes for older people with cancer, as well as analysing key features of the workforce associated with those improvements.

## METHODS

Systematic methods were used to guide searching, selection and analysis.[18] Searches for primary research evaluating workforce interventions for older people with cancer were undertaken in August 2016. Studies were identified by searching electronic databases, scanning reference lists of articles and by contacting study authors. A detailed search strategy was tested in MEDLINE (table 1). The search was additionally tailored for database-specific subject headings and applied in: PsycINFO, Cumulative Index to Nursing and Allied Health Literature (CINAHL), Allied and Complementary Medicine Database (AMED), Embase, Web of Science, Cochrane Central Register of Controlled Trials (CENTRAL), AgeInfo and Scopus (see online supplementary file 1). Searches were limited to the English language. No date limit was applied to ensure a comprehensive overview of developments in the field. The PRISMA (Preferred Reporting Items for Systematic Reviews and Meta-Analyses) guidelines have been used to guide reporting (see online supplementary file 2).[19]

Eligible study types included randomised controlled trials (RCTs), quasiexperimental or observational studies with a clearly defined workforce variable or intervention with a comparison between different exposure levels, and qualitative studies evaluating features of the workforce from the perspective of older people with cancer and where the role of the workforce forms a central part of the research question. We defined workforce-based interventions as any intervention where the main mode of action was through a change in the composition, roles, knowledge, skills or attitudes

of individuals or groups in a care delivery role, paid or unpaid, not including family or informal caregivers. Papers included reported on studies conducted with participants identified as older people (age 60+) at any stage in the cancer journey (from targeted screening through to end of life). Papers included reported on either:

► outcomes of interventions to improve the knowledge, attitudes or skills of the workforce delivering cancer care and treatment to older people;

► outcomes of interventions involving a change in the composition and/or skill mix of the workforce delivering cancer care for older people including (but not limited to) role substitution, new roles or adding specialist practitioners to the team;

► outcomes of interventions routinely targeted at older people with cancer, which were reported to require significant workforce reconfiguration or the implementation of new roles;

► associations between the composition and characteristics of the cancer care workforce (including, but not limited to, staffing levels, skill mix, training, knowledge attitudes and skill) and outcomes for older people with cancer.

Studies reporting solely on drug, treatment or other therapeutic interventions (without specific focus on the workforce delivering those interventions) were not included.

Titles and abstracts from the searches were screened against the inclusion criteria by GL to exclude irrelevant papers. Five per cent of titles/abstracts were also independently reviewed by another team member (JB, PG or TW) to confirm exclusion decisions. Full-text papers were retrieved for all papers that screened positively against inclusion criteria or about which a clear decision could not be taken (due to lack of information). Each full-text paper was reviewed independently by two team members followed by a decision to include or exclude. These reviews were followed by further team discussion to finalise inclusion. The search and selection process is summarised in the PRISMA flow chart (figure 1).[19]

Data on aim, design, setting, sample, intervention, outcome and results were extracted systematically from eligible papers using data extraction tables developed by the team (see online supplementary file 3). We adapted the GRADE (Grading of Recommendations Assessment, Development and Evaluation) system as used by Cochrane for rating evidence[18] to guide a broad assessment of individual study quality and thereby the contribution studies made to the review. Initial quality ratings based on study design were upgraded or downgraded depending on presence of factors considered to strengthen or weaken the evidence. Two members of the team independently reviewed all included papers. Discrepancies were discussed and ratings confirmed through discussions involving both raters and a third team member. No studies were excluded based on this

| Table 1 | Example of search strategy for MEDLINE (EBSCOHOST) | |
|---|---|---|
| **Concept 1** | **Concept 2** | **Concept 3** |
| 1. TI Elderly OR AB Elderly | 10. TI Cancer OR AB Cancer | 14. TI Workforce OR AB Workforce |
| 2. TI Geriatric* OR AB Geriatric* | 11. TI Oncolog* OR AB Oncolog* | 15. TI 'Health professionals' OR AB 'Health professionals' |
| 3. TI 'Older people' OR AB 'Older people' | 12. MM Neoplasms | 16. TI 'Healthcare professionals' OR AB 'Healthcare professionals' |
| 4. TI 'Older patient*' OR AB 'Older patient*' | 13. 10 or 11 or 12 | 17. TI 'Health care professionals' OR AB 'Health care professionals' |
| 5. TI 'Older person' OR AB 'Older person' | | 18. TI 'Health personnel' OR AB 'Health personnel' |
| 6. TI 'Older adult*' OR AB 'Older adult*' | | 19. TI 'Healthcare personnel' OR AB 'Healthcare personnel' |
| 7. MM Aged | | 20. TI 'Health care personnel' OR AB 'Health care personnel' |
| 8. MM Frail Elderly | | 21. TI 'Medical personnel' OR AB 'Medical personnel' |
| 9. 1 or 2 or 3 or 4 or 5 or 6 or 7 or 8 | | 22. TI 'Advanced Practice nurse' OR AB 'Advanced Practice Nurse' |
| | | 23. TI 'Clinical nurse specialist' OR AB 'Clinical nurse specialist' |
| | | 24. TI Geriatrician* OR AB Geriatrician* |
| | | 25. TI Gerontologist* OR AB Gerontologist* |
| | | 26. TI 'Allied health professionals' OR AB 'Allied health professionals' |
| | | 27. TI Training |
| | | 28. TI Educat* |
| | | 29. TI 'Skill mix' OR AB 'Skill mix' |
| | | 30. TI 'Grade mix' OR AB 'Grade mix' |
| | | 31. TI 'Staff development' OR AB 'Staff development' |
| | | 32. TI Staff* W1 level* OR AB Staff* W1 level* |
| | | 33. TI Teamwork OR AB Teamwork |
| | | 34. MM Health manpower |
| | | 35. MM Health personnel |
| | | 36. MM Attitude of Health personnel |
| | | 37. MM Professional Competence |
| | | 38. MM Staff development |
| | | 39. MM Education, professional |
| | | 40. MM Nurse's role |
| | | 41. MM Geriatric assessment |
| | | 42. MM Health services for the aged |
| | | 43. or/14–42 |
| | | 44. 9 AND 13 AND 43 |
| | | 45. English language filter |

assessment but lower quality studies were given less weight in the analysis.

Due to the heterogeneity of interventions and outcomes, a narrative analysis of study findings was merited.[20] Studies were grouped around the patient or service problems the interventions were targeting. Results were tabulated and the findings of effectiveness of individual interventions were plotted within these groups and used as the basis for an analysis of the strength of evidence of effectiveness across these groups and the field as a whole. We recorded and tabulated both the direction of differences between groups (where reported) and statistical significance of differences. Due to the number of different outcomes across the 24 studies, we report, within the Results section, for the primary outcomes where there is evidence of significant differences between groups, rather than narrating the full set of results for each individual paper. A review protocol is available from the study team on request.

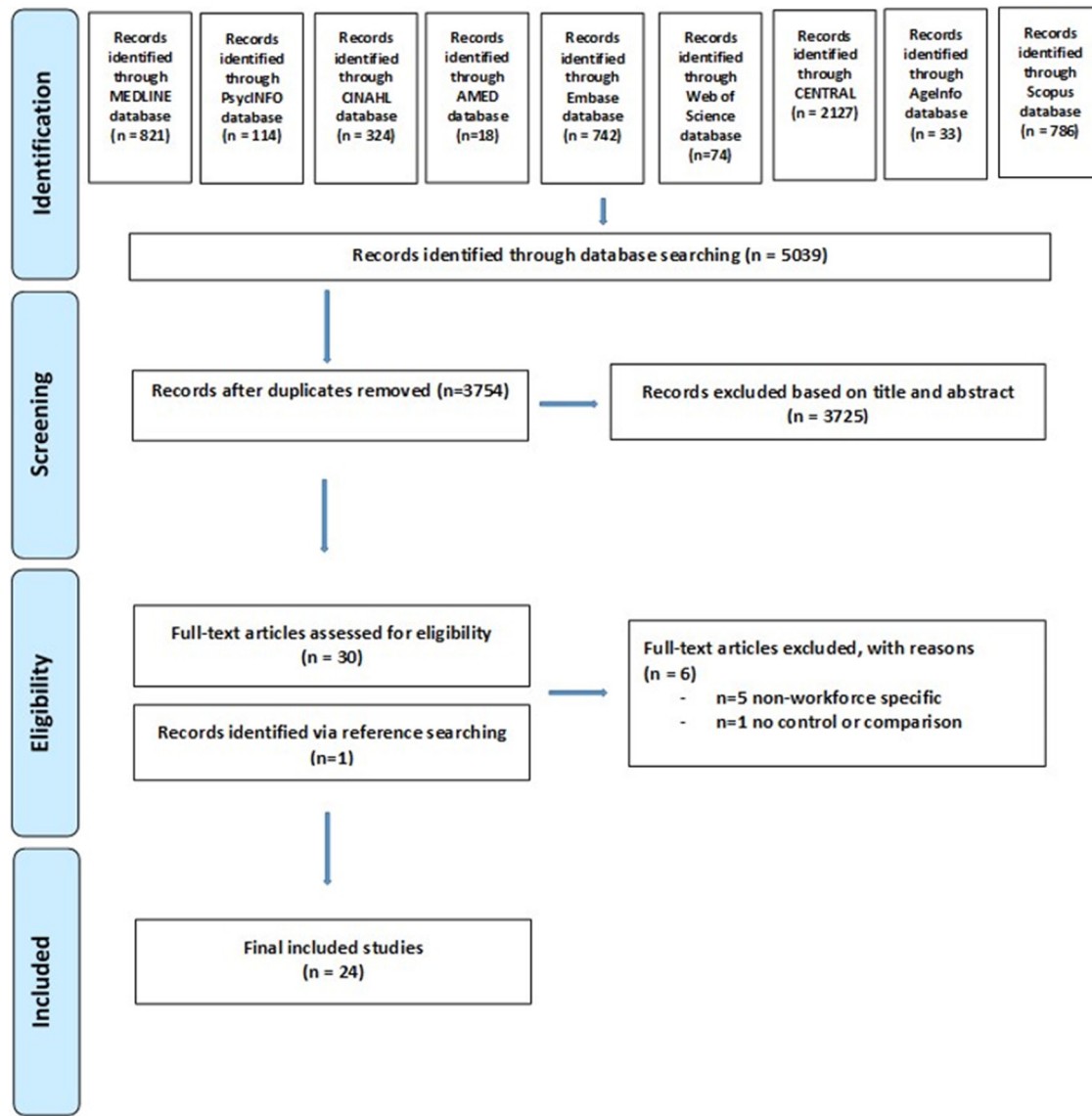

**Figure 1** PRISMA (Preferred Reporting Items for Systematic Reviews and Meta-Analyses) study selection flow chart.

## RESULTS

We identified 24 eligible published journal papers (23 quantitative and 1 qualitative study) covering 22 interventions and reporting on 4555 patient participants age 60+ from targeted screening, through cancer diagnosis and treatment and beyond. All but one study were conducted in USA or Europe. The studies report on 27 individual primary outcomes and 42 individual secondary outcomes (using a range of measures) across the studies corresponding to 41 different outcomes in total (n=38 of these were patient-related outcomes and the other 3 outcomes were focused directly on the workforce). As detailed below and illustrated in table 2, 17 studies were assessed as low or very low quality, with 4 studies rated as medium and 3 as high quality.

The point of the cancer journey each intervention was targeted at varied widely. Interventions ranged from targeted screening stage (n=1) and from diagnosis (n=4); to treatment phase/hospital stay (n=11); to those primarily focused on patients after the completion of their treatment (n=6); hospice care (n=1) or home care for patients with advanced cancer (n=1). The majority of the interventions were limited to specific tumour types: 15 involved participants with a range of cancer types, but some involved more homogeneous populations: 6 were for patients with breast cancer, 1 intervention targeted patients with prostate cancer, another involved those with gastrointestinal cancers and 1 was aimed at breast and cervical screening.

Only two interventions were directly targeted at improving the knowledge, attitudes or skills of the workforce delivering cancer care and treatment to older people through training[21 22] and only two studies directly addressed the second objective of the review to assess the salient features of the cancer care workforce: one qualitative study considered the features of the nursing workforce which older patients felt were important in their care[23] and one study looked at the impact of

**Table 2** Summary of studies included in review

| Source; study design; final quality rating | Setting and sample | Intervention and workforce | Primary outcomes (secondary outcomes) | Results |
|---|---|---|---|---|
| **Regular and timely access to care and treatment** | | | | |
| Basu et al[25] Observational: care control ++ | Women aged 61+ with breast cancer, n=86 One cancer centre, USA | I: Patient navigation: support and coordination of patient care T: From point of diagnosis to survivorship clinic C: No navigation W: Breast cancer nurse | Time from diagnosis to oncology appointment | Time to consultation decreased by 4.9 days (p=0.0002) |
| Goodwin et al[27] RCT ++++ | Women aged 65+ newly diagnosed with breast cancer, n=335 13 community and 2 public hospitals, USA | I: Case management: nurse as educator, counsellor, advocate and care coordinator T: 12 months service C: Usual care (unclear) W: Nurse case manager | Treatment received in 6 months after breast cancer diagnosis (patient satisfaction; arm function) | More intervention women saw radiation oncologist (36% vs 19.3%) (p=0.006), received more breast-conserving surgery (28.6% vs 18.7%; p=0.031) and radiation therapy (36.0% vs 19.0%; p=0.003), had more breast reconstruction surgery (9.3% vs 2.6%, p=0.054); reported choice in treatment (82.2% vs 69.9%, p=0.020) No differences between groups in percentages who saw an oncologist, discussed breast reconstruction, underwent complete surgical staging, or tissue sent for hormone receptor assay |
| Mandelblatt et al[35] Controlled before and after study ++ | Women aged 65+ screening for breast or cervical cancer, n=673 Two public hospitals, USA | I: Screening intervention during routine visits T: At scheduled appointments C: Physician reminder system W: Nurse practitioner | Annual screening rates for Pap tests and mammographies | Annual intervention site Pap test rate increased (17.8%–56.9%), and mammographies (18.3%–40%) compared with control site increase of Pap test rate from 11.8% to 18.2% and no change (18%) for mammography (p=0.01) |
| Somana-Ehrminger et al[32] Observational ++ | Women aged 75+ with breast cancer, n=206 Breast and gynaecological cancer registry, France | I: Geriatrician referral and treatment plan C: Patients with no GOC W: Geriatrician, dietitian, psychologist, physical therapist or social worker | Independent impact of GOC | GOC patients more likely to receive mastectomy and adjuvant therapy (p<0.0001); and less likely to be treated by breast-conserving surgery and adjuvant therapy (p=0.003) |
| **Complications and specific problems of cancer treatment** | | | | |
| Bourdel-Marchasson et al[34] RCT ++ | Chemotherapy patients aged 70+, n=336 12 public and private settings, France | I: Face-to-face dietary counselling T: 6 visits (3–6 months) C: Usual care W: Dietitian | 1-year mortality (chemotherapy management; unplanned hospitalisation; 2-year mortality) | Difference of 178 kcal/day dietary intake in intervention group (p<0.01) No difference in other outcomes |
| Hempenius et al[30] RCT ++ | Frail adults aged 65+, elective surgery for solid tumour, n=260 Teaching hospital and community hospital, Netherlands | I: Delirium prevention: assessment, monitoring and individualised treatment plan T: During hospital stay C: Usual care. Additional geriatric care provided on referral W: Geriatric team supervised by a geriatrician | Incidence of postoperative delirium up to 10 days (severity of delirium; length of hospital stay; complications; mortality; care dependency; QOL) | Significant difference in return to preoperative living situation (67.3% vs 79.1%, OR: 1.84, 95% CI 1.01 to 3.37) No significant difference in other outcomes |
| Kalsi et al[44] Observational ++ | Adults aged 70+ with cancer, n=135 One hospital, UK | I: Geriatrician CGA and intervention plan for identified need T: Prechemotherapy and further support as needed C: Standard oncology care W: Geriatrician | CGA impact on chemotherapy tolerance and toxicity; rate of planned completion of cancer (treatment modifications; early treatment discontinuation; death at 6 months) | Intervention more likely to complete planned cancer treatment (33.8% vs 11.4%, OR 4.14 (95% CI 1.50 to 11.42), p=0.006) and fewer required treatment modifications (43.1% vs 68.6%, OR 0.34 (95% CI 0.16 to 0.73), p=0.006) Lower toxicity rate in intervention group (43.8% vs 52.9%, p=0.292) No differences in death rates |

Continued

**Table 2** Continued

| Source; study design; final quality rating | Setting and sample | Intervention and workforce | Primary outcomes (secondary outcomes) | Results |
|---|---|---|---|---|
| McCorkle et al[37] RCT ++++ | Adults aged 60+ with postsurgical cancer, n=375 Comprehensive cancer centre, USA | I: Specialised home care APNs assess and monitor physical, emotional and functional status of patients, provide direct care, access services and other resources from the community, and provide teaching, counselling and support during recovery T: 4 weeks with three home visits and five telephone contacts C: Usual follow-up care in an ambulatory setting and routine outpatient follow-up W: APNs | Length of survival (depressive symptoms; symptom distress; functional status) | Late-stage patients, improved 2-year survival in intervention group: 66.7% vs 39.6% (p<0.05). No difference for early-stage patients Risk of death higher for control (adjusted HR 2.04; 95% CI 1.33 to 3.12; p=0.001) compared with intervention group No differences between groups in other outcomes |
| **Comorbidities and complex health needs** | | | | |
| Deliens et al.[33] Uncontrolled before and after study + | Adults aged 70+ with cancer (non-haematological) hospitalised, n=91 Geriatric oncology unit, tertiary hospital, Belgium | I: Medication review: identification of PIMs and drug interactions T: From point of admission and during hospitalisation C: Before to after W: Clinical pharmacist | PIMs using START and STOPP criteria (drug-to-drug interactions) | START criteria: 41 PIMs for 31 patients (34%) at hospital admission compared with 7 PIMs for 6 persons (7%) at discharge STOPP criteria: 50 PIMs for 29 patients (32%) at admission compared with 16 PIMs for 14 persons (16%) at discharge |
| Fann et al[26] RCT +++ | Adults aged 60+ with diagnosis of non-skin cancer and major depression or dysthymia, n=215 18 primary care clinics at 8 diverse healthcare organisations, USA | I: Depression management: education, 'behavioural activation', treatment support. T: Up to 12months. Follow-up usual care 12months more C: Usual care: received routinely available depression treatment W: Depression care manager (nurse or clinical psychologist) collaborative with primary care | Depression treatment response (health-related QOL; health-related impairments: work, family, social functioning; patient satisfaction) | Intervention twice as likely to experience a depression treatment response at 12months than control (39% vs 20%; p=0.029) and at 18months (38% vs 16%; p=0.012) Remission rates higher in intervention group versus control group at 6months (32% vs 16%, p=0.006) and 12months (22% vs 9%, p=0.031) Less functional impairment at 12months (p=0.011) and greater QOL (p=0.039) |
| Herr et al[22] RCT (cluster) +++ | Adults aged 65+ with cancer receiving hospice care, n=738 Staff: nurses (n=383 pre, n=415 post) and physicians (n=16) 16 hospices, USA | I: Workforce: to promote adoption of evidence-based pain practices. Included: training, assessment of data, champion input, senior leadership engagement T: Engagement phase 5months, 12-month intervention C: Hospices received clinical practice guidelines W: 3 days training. Selection of local pain facilitators, nurse and physician champions, grant expert nurse input, nurse and physician champion | Workforce: adoption of evidence-based cancer pain practices (pain severity) | No significant difference in improvement on cancer pain practice index between intervention and control Decrease in patient pain severity from pre to post in intervention group greater (p=0.1032) |

Continued

**Table 2** Continued

| Source; study design; final quality rating | Setting and sample | Intervention and workforce | Primary outcomes (secondary outcomes) | Results |
|---|---|---|---|---|
| Johansson et al[41] RCT ++ | Adults aged 70+ newly diagnosed with prostate, GI or breast cancer, n=161 Primary healthcare services, Sweden (other participants reported: n=255 under 70 years) | I: Intensified primary healthcare. Individual support: nurse support, nutritional support and individual psychological support. T: Starting from diagnosis C: Standard care + group rehabilitation W: Home care nurse, dietitian and psychologist. GPs and nurses trained in pain, nausea and diet in final-stage life | Utilisation of specialist care | Mean days of hospitalisation for older intervention patients than control (3.8 vs 8.9, p<0.01) 4 of 82 older intervention patients admitted compared with 12 of 79 older control patients (p<0.05) 10 out of 82 made acute visits to outpatient clinics compared with 22 of 79 in control group (p<0.05) |
| Rao et al[31] RCT +++ | Adults aged 65+ with cancer, frail and hospitalised, n=99 11 medical centres, USA | I: Assessment and monitoring by geriatric team: (1) geriatric inpatient + usual outpatient; (2) usual inpatient + geriatric outpatient; (3) geriatric inpatient and outpatient T: 1-year study C: Usual care: all hospital services except from geriatric team W: Core team: geriatric medicine attending physician, fellow or intern, a nurse practitioner, social worker | Survival: health-related QOL (functional status; physical performance) | No difference in survival for patients with cancer regardless of treatment group Significant effect of geriatric inpatient care versus usual inpatient care: mean change in score from randomisation to discharge: bodily pain (28.7 vs 10.1), p=0.09; emotional limitation (29.3 vs 2.7), p=0.01. Effect on bodily pain sustained at 1 year (37.6 vs 9.9) No effect of geriatric outpatient care on any of the QOL parameters No effect of either inpatient or outpatient geriatric care on the functional status of patients with cancer |

### QOL, physical and psychological functioning

| Source; study design; final quality rating | Setting and sample | Intervention and workforce | Primary outcomes (secondary outcomes) | Results |
|---|---|---|---|---|
| Chock et al[40] (secondary analysis of Clark et al[66]) RCT ++ | Adults aged 65+ with advanced cancer treated with radiotherapy, n=16 Cancer centre, USA (other participants reported: n=38 under 65 years) | I: QOL intervention with telephone follow-up: physical therapy, education, cognitive behavioural interventions, discussion and support, spiritual reflection and relaxation training T: 6 sessions 90 min, 2–4 weeks and 10 brief structured telephone sessions C: Standard care W: Multidisciplinary (including physical therapist, clinical psychologist, APN, chaplain) | QOL; mood | Significant difference at week 4 only in mean overall QOL older versus younger adults (74.4 vs 62.9, p=0.040) Significantly lower anger-hostility dimension of mood measure at all weeks for older versus younger patients. Week 4: 95.0 vs 86.4, p=0.028; week 27: 92.2 vs 84.2, p=0.027; week 52: 96.3 vs 85.9, p=0.005 No other significant differences |
| Heidrich et al[38] Two pilot RCTs and one observational study ++ | Women aged 65+, 1 year postdiagnosis of non-metastatic breast cancer, n=82 (total) Oncology clinics, cancer centre, USA | I: Pilot 1—symptom management (IRIS): counselling interview and telephone follow-up on symptom management at 4 weeks; pilot 2—addition of four biweekly telephone reinforcement sessions; pilot 3—intervention by phone only C: (1) usual care; (2) delayed IRIS (waitlist) control; (3) no control (IRIS group only) T: 4 weeks (pilot 1) W: APN | Feasibility, acceptability (symptom distress; symptom management; QOL; mood; barriers to symptom management; communication difficulty) | Feasibility: across all studies, 76% of eligible women participated, 95% completed the study, 88% reported the study was helpful and 91% were satisfied with the study. Pilot 1: no significant difference in symptom distress. Significant decrease in distress baseline to follow-up in intervention group; significantly more women in intervention reported changing self-care of symptoms (p<0.05); no significance QOL differences. Pilot 2: significant less symptom duration compared with control at 8 weeks (p<0.01). At 16 weeks, intervention group more likely to have talked to healthcare provider, begun new symptom treatment and changed self-care symptoms (p<0.05). No significant QOL differences. Negative attitudes from healthcare providers reported by 5%–20% of women and communication difficulties by 5%–45% of women. Pilot 3: No significant differences (no control) from baseline to 8 weeks. Symptom interference decreased (and negative mood from symptoms). Symptom duration interference and negative mood from symptoms decreased. No QOL change |

Continued

**Table 2** Continued

| Source; study design; final quality rating | Setting and sample | Intervention and workforce | Primary outcomes (secondary outcomes) | Results |
|---|---|---|---|---|
| Kornblith et al[28]<br>RCT<br>++ | Adults aged 65+ with breast, colon or prostate cancer, n=131<br>Cancer centres/university settings, USA | I: Telephone monitoring of distress providing support (plus educational materials)<br>T: Over 6 months—monthly monitoring<br>C: Educational materials alone, referred to oncology nurse upon evaluation if distressed significantly<br>W: Trained graduates monitoring telephone calls. Referral onto an oncology nurse where indicated | Psychological distress | Lower anxiety and depression mean HADS total score for intervention 6.01 vs 8.20 control (p<0.0001); HADS depression subscale, intervention 3.20 vs 4.08 control (p=0.0004); HADS anxiety subscale intervention 2.81 vs 3.25 control (p<0.0001), at 6 months controlling for study entry levels<br>No differences on other measures of psychological distress |
| Lapid et al[42]<br>Secondary age group analysis of Rummans et al[67]<br>RCT<br>++ | Adults aged 65+ newly diagnosed with advanced cancer, n=33<br>Cancer centre, USA | I: Multidisciplinary psychosocial QOL sessions<br>T: Eight 90 min sessions, 4 weeks after enrolment<br>C: Standard care (regular outpatient visits with oncologist and allied healthcare providers)<br>W: Led by psychiatrist or psychologist and cofacilitated by a nurse, physical therapist, chaplain or social worker. Leaders trained in materials and observed sessions | QOL | Higher overall QOL intervention group scores throughout the study, not significant<br>Higher QOL scores at week 4 intervention versus control (79.3 vs 62.9, p=0.0461)<br>Improvement in QOL scores for intervention at weeks 4 and 8 compared with older control group |
| Mantovani et al[29]<br>RCT<br>++ | Adults aged 65+ with cancer, n=72<br>Inpatient setting at medical oncology clinic, Italy | I1. Emotional and practical support from volunteers and I2. with structured psychotherapy<br>T: Weekly sessions of 1 hour for 6 months<br>C: Pharmacological only<br>W: Trained volunteers | QOL | Non-significant between group differences in functional status/physical symptom improvements over time: Karnofsky's Performance Status Scale (F=9.90, 2 df, p<0.001).<br>No differences on Spitzer's QOL Index or Functional Living Index—within/between groups<br>Significant between group differences: State-Trait Anxiety Inventory control significantly worsened and intervention groups significantly improved (I1 improved more than I2) (F=4.50, 2 df, p<0.05)<br>Beck Depression Inventory: control group unchanged, both intervention groups improved (F=229.66, 2 df, p<0.01) |
| Sajid et al[43]<br>RCT (pilot)<br>++ | Men aged 70+ with prostate cancer and hormone therapy, n=19<br>Two medical oncology clinics, USA | I1. EXCAP (home-based walking and resistance intervention)<br>I2. Technology-mediated walking and resistance intervention using Wii Fit<br>T: One face-to-face session then 6–12 weeks home based<br>C: Usual care<br>W: Trained exercise physiologist | Functional and aerobic (skeletal muscle and muscular mass measure; handgrip strength; chest repetition test; DEXA scan) | EXCAP intervention arm higher rate of change in steps per day at each follow-up (+2720 steps) (p<0.01) compared with control (+97 steps) and Wii Fit arm (+382 non-significant)<br>EXCAP arm had a 2.3 point change in physical battery score after 12 weeks, compared with 0.6 points in the Wii Fit arm and −0.5 points in the usual care arm<br>No other significant differences in outcomes |
| Suh et al[39]<br>RCT<br>+++ | Adults aged 65+ completed active treatment for gastrointestinal cancers, n=63<br>Cancer centre, South Korea | I: 8 weeks of Qi exercise and 1 hour face-to-face counselling on physical and psychological factors<br>T: 8 weeks<br>C: Usual care<br>W: Two Qi exercise trainers, APNs | Physical activity (BMI; body weight; nutritional status; symptom experience; self-efficacy; self-esteem) | Physical activity increased in both groups, extent of increase greater in intervention group (p=0.005) Difference in amount of exercise over time between groups (p=0.002)<br>No between-group difference in BMI<br>Nutritional status in both groups improved over time. The degree of reduction, however, was significantly larger in intervention group (p=0.048), and same in interaction between group and time<br>Both group and interaction factors have significant positive difference in symptom experience, health promotion and self-esteem for intervention |

Continued

**Table 2** Continued

| Source; study design; final quality rating | Setting and sample | Intervention and workforce | Primary outcomes (secondary outcomes) | Results |
|---|---|---|---|---|
| Yagli and Ulger[36] Controlled before and after study ++ | Women aged 65–70, 6 months after chemotherapy for breast cancer, n=20 Department of physiotherapy and rehabilitation, Turkey | I: 8 sessions of 1-hour yoga classes T: 8 weeks C: Exercise programme for 8weeks W: Existing physiotherapist (yoga teacher) | QOL; depression levels; levels of pain, fatigue and sleep quality | All patients' QOL scores improved pre to postyoga and exercise interventions Total scores and some subcategories of the Nottingham Health Profile showed significant difference in favour of the yoga group (p<0.05) but not on energy level and pain where there were no differences Significant better fatigue and sleep quality in yoga group postintervention (p<0.05) |

Communication between patients and healthcare professionals

| Source; study design; final quality rating | Setting and sample | Intervention and workforce | Primary outcomes (secondary outcomes) | Results |
|---|---|---|---|---|
| Devik et al[23] Qualitative ++ | Adults aged 65+ with advanced cancer, n=9 Patients' homes in rural Norway | I: Qualitative study of home nursing care to patients with advanced cancer in rural locations C: NA W: District nurses in normal role | Patient experience | Importance of nurses having a person-centred manner Ability to show a genuine and empathic interest in the patients Technical skills or special competences less discussed than personal qualities, such as having a sense of humour or generosity Good listening and communication skills |
| van Weert et al[21] RCT (cluster) ++++ | Adults aged 65+ with cancer receiving chemotherapy, n=210 Staff: oncology-trained nurses, n=77 12 wards of 10 hospitals, Netherlands | I: Workforce: communication skills training in delivery of chemotherapy education to patients T: 3-month implementation C: Nurses continued to provide patient education as usual W: Nursing and specialised oncology nursing roles | Effects on quality of staff communication; effects on content of the consultation (patient recall of information) | Significant improvement in discussing realistic expectations. C: -0.20; I: 0.45 (total between-group difference 0.65) (p<0.01) Significant decrease in rehabilitation information pre to postchange. C: 0.08; I: -0.38 (total between-group difference -0.45) (p<0.01) No significant changes in categories treatment-related information and coping information discussed Non-significant: intervention group showed significant decrease in number of items discussed Less history taking pre to post (C: 1.83; I: -2.33; between-group difference -4.17; p<0.001) and less talking about all different side effects pre to postchange (C: 1.98; I: -5.71; total 7.68; p<0.001) Patients in intervention asked more questions (M=10.76) than control (M=6.69; p<0.05) Marginal significance for intervention group: proportion recall of recommendations post versus pre (C: -3.34; I: 6.39; total: 9.73; p<0.10) |
| Yeom and Heidrich[24] Observational ++ | 190 women at least 1 year postbreast cancer diagnosis, n=190 Community, an oncology clinic and a state tumour registry, USA | I: Symptom management (IRIS): counselling interview and telephone follow-up on symptom management T: 8-week intervention with 16-week follow-up point in the RCT C: Waitlist control subjects offered intervention after 16-week follow-up assessment W: APN | Negative beliefs about symptom management (QOL; purpose in life; positive relations with others) | Significant direct effects on SMBQ (p<0.00) and Communication Attitudes Questionnaire (p=.012) or Communication Difficulties Questionnaire Communication difficulties significant direct, negative effects on all four dimensions of QOL Significant total effects of SMBQ on MCS (mental QOL) (p=0.001) and PIL and PR (p<0.001) but not physical component (PCS). SMBQ predicted lower levels of QOL in three of four dimensions None of the four indirect effects of SMBQ on QOL through CommD was significant, indicating that CommD does not mediate the effects of SMBQ on QOL The total effects of CommA on four QOL measures were not significant. However, the indirect effects for MCS (p=0.05), PIL (p<0.05) and PR (p<0.05) through CommD were significant, indicating that CommD mediates the effects of CommA on MCS, PIL and PR |

Quality ratings: high ++++; moderate +++; low ++; very low + intervention/workforce description.
APN, advanced practice nurse; BMI, body mass index; C, control group; CGA, comprehensive geriatric assessment; df, degrees of freedom; DEXA, dual-energy X-ray absorptiometry; EXCAP home-based walking and resistance intervention; GI gastrointestinal; GOC, geriatric oncology consultation; GP, general practitioner; HADS Hospital Anxiety and Depression Scale; I, intervention; IRIS individualized representational intervention to improve symptom management; MCS, mental component summary; NA, not applicable; PCS, physical component summary; PIL, purpose in life; PIM, potentially inappropriate medications; PR, personal relations; QOL, quality of life; RCT, randomised controlled trial; SMBQ, Symptom Management Beliefs Questionnaire; START, screening tool to alert doctors to right treatment; STOPP, screening tool of older person's potentially inappropriate prescriptions; T, time point; W, workforce involved.

healthcare professionals communication on participants' views about their symptom management.[24] The remaining studies reported on improving older people's outcomes via interventions involving a change in the workforce. In five interventions new roles were tested: nurse navigator,[25] depression care manager,[26] nurse case manager,[27] telephone support (trained graduates)[28] and social support volunteers.[29] In other studies, support from additional workforce members was provided to patients. Four studies reported on the increased involvement of a geriatrician or a geriatrics team,[27 30–32] one reported on the input of a clinical pharmacist[33] and one study reported on the input of an additional dietitian.[34] In two studies, a current staff member had a different function; in one study a nurse provided targeted cancer screening[35] and in another study a physiotherapist designed exercise and yoga programmes.[36] Three interventions used advanced practice nurses (APN)—one in a home care capacity[37] and two in counselling roles.[38 39] In three studies, the role of multidisciplinary team members was highlighted.[40–42] In some papers, although a named member or members of the workforce were reported to have implemented or carried out the intervention, it was unclear as to the exact nature of their position. This was the case with two studies using exercise physiologists where it could not be determined if they were existing or new staff members.[39 43] Only seven studies referred to an explicit theoretical framework or model in intervention design.[21 22 24–26 38 39]

Because of the heterogeneity of studies retrieved (and the small number of studies that addressed the review's second objective), we reviewed evidence of the effectiveness of interventions by study type established through particular problems (related to older people with cancer) that the respective interventions were addressing and, subsequently, ways in which workforce requirements were being adapted to meet needs and improve outcomes related to these patient problems. The results in table 2 and set out below are displayed using these individual types.

### Regular and timely access to care and treatment

Four studies focused on interventions targeted at the problem of systemic delays or inequitable access to treatment in the cancer journey for older people. They provide some promising evidence that providing additional support to some groups of older patients with cancer can help them navigate the system and access treatment thereby improving the speed and efficacy of care. However, three of these papers provide only low-quality evidence.

A high-quality RCT reported that older women with breast cancer in the care of a nurse case manager acting as an educator, counsellor and coordinator were significantly more likely to see a radiation oncologist as part of initial evaluation, and to receive breast-conserving surgery and radiation therapy.[27] Further, the difference in receipt of appropriate treatment between women with characteristics associated with lower rates of appropriate

treatment (75+, being unmarried, living alone and being a member of an ethnic minority group) and their respective comparison groups were diminished or eliminated in the intervention group. An observational study reported that a breast cancer nurse navigator providing support and coordination of patient care from diagnosis until entry into survivorship clinic significantly shortened time to consultation for patients aged 61+ years.[25] A nurse practitioner role was used in a quasiexperimental study to improve screening rates for older Black women of low socioeconomic status by offering screening during a routine visit.[35] Nurse practitioner follow-up screening rates were significantly higher than baseline, compared with control group follow-up rates. A further study assessed the impact of a geriatrician consultation and treatment plan through an analysis of registry data of older patients with breast cancer.[32] Patients who had a consultation had more comorbidities and more advanced and aggressive tumours, were more likely to receive mastectomy and adjuvant therapy, and were less likely to be treated by breast-conserving surgery and adjuvant therapy.

### Complications and specific problems from cancer treatment

Four studies reported the use of workforce members with specialist skills to address cancer treatment complications and impact on mortality and survival. None of the three low-quality studies found any intervention effect on mortality rates, but the one high-quality RCT found that specialised home care APN (used to enhance surgical recovery) increased 2-year survival for patients with late-stage cancer in the intervention group.[37]

Other lower quality studies in this group included evaluations of face-to-face counselling to address nutritional intake for patients treated with chemotherapy and at risk of malnutrition,[34] an intervention focused on the prevention of postoperative delirium with input from a geriatric team[30] and comprehensive geriatric assessment (CGA) targeted at chemotherapy tolerance and toxicity.[44] The observational study evaluating CGA for older chemotherapy patients found that CGA patients were more likely to complete cancer treatment as planned but no significant differences were found in relation to mortality or other outcome measures in relation to the interventions in any of these three studies.

### Comorbidities and complex health needs

The five studies reported here target the health issues that may accompany a cancer diagnosis, but also broader health problems that may not directly relate to the cancer. They highlight the importance of recognising and addressing these needs, although the range of outcomes and the variable quality of evidence (three studies of medium quality; two were low quality) make it difficult to draw firm conclusions about the best use of workforce support in this sizeable area.

A cluster RCT evaluating a hospice staff training programme on improving pain assessment and management did not find significant practice improvements or decreases in patient pain severity associated with the

intervention.[22] In a different study, a secondary analysis of RCT data on the impact of a depression care manager providing education and support for older patients with depression found that intervention patients with a cancer diagnosis were twice as likely to experience a depression treatment response at 12 months compared with usual care.[26] Rao et al also reviewed the outcomes for patients with cancer from a wider RCT evaluating the impact of involving a geriatric team in the care of inpatients and outpatients diagnosed with frailty.[31] The inpatient intervention group showed significant improvements in bodily pain and mental health versus the usual inpatient care group but there was no impact on survival rates. There were no intervention effects on outpatients. An uncontrolled before and after study reported that using a clinical pharmacist to identify patients' potentially inappropriate medications (PIMs) reduced the number of PIMs at discharge versus admission.[33] A low-quality RCT reported that intensified primary healthcare support significantly reduced the number of days in hospital for an intervention group of patients with advanced cancer compared with patients receiving standard care.[41]

### QOL, physical and psychological functioning

Eight studies focused on addressing QOL across its physical and psychological aspects. This group of interventions used a range of workforce members (often in therapeutic or supportive roles) from physiotherapists to APN to trained voluntary input, to address a range of factors underpinning QOL. They showed mixed evidence of effectiveness. Seven of the studies in this group provided low-quality evidence.

Three studies focused on physical functioning in particular. In an RCT with low recruitment rate and possible selection bias, exercise physicians provided Qi exercise training.[39] Both usual care and intervention participants increased their activity levels but the extent of the increase was significantly greater in the intervention group. The intervention also used APN delivering face-to-face counselling and significant improvements in symptom experience, self-efficacy and self-esteem were reported. A controlled before and after study compared the effect of yoga classes (with the input of a physiotherapist/yoga teacher) with a standard exercise programme.[36]

QOL scores after the programme were better than before for both groups, but some QOL parameters improved more for those included in the yoga intervention. A pilot RCT with small sample and high dropout compared two exercise forms implemented by a physiologist (compared with usual care) and found significant activity increases for the group using a home-based walking and resistance intervention.[43]

Two similar interventions involved a multidisciplinary team approach for a range of QOL domains; however, both of these secondary analyses reported on very small sample sizes of older adults within wider QOL interventions. Lapid et al[42] found in a secondary analysis of a small sample of patients in a wider RCT, that higher QOL scores were reported for older patients who received multidisciplinary emotional and practical support. However, in the study by Chock et al,[40] the authors did not find any lasting differences on QOL for older intervention participants against their younger counterparts, apart from an improvement in anger-hostility.

APNs were used in a symptom management intervention in the two pilot RCTs and the observational study reported by Heidrich et al.[38] Some evidence of effectiveness was reported for improving self-care and reducing symptom distress and duration, but there was no impact on QOL.

Two studies used trained volunteers to bolster psychological support. A secondary analysis of RCT data was used to evaluate the effect of using trained graduate support workers to provide initial distress monitoring to patients over the telephone.[28] Intervention patients had significantly lower anxiety and depression at 6 months than patients receiving educational materials alone. However, no other differences in psychological well-being were detected. Mantovani et al[29] also used trained support volunteers to provide emotional and practical support. An RCT with small sample size was used to compare this support with pharmacological treatment alone, and further with the addition of psychotherapy. Significant improvements in anxiety and depression were reported for the groups receiving voluntary support and/or additional psychotherapy. However, there were no significant differences on other QOL measures.

### Communication between healthcare professionals and older people with cancer

Three studies focused on addressing the communication needs of older people with cancer. One high-quality study offered communication skills training to staff with varied success[21] and the other two low-quality studies highlighted the importance of good communication as a prerequisite for cancer nurses related to improving older patients' QOL.

A cluster RCT found that training nursing staff to improve chemotherapy patient education led to a significant, positive effect for 'discussing realistic expectations.'[21] Significantly less history taking was also observed pre to post in the intervention group, as well as less talking about all the possible side effects; both points of attention during training. No other significant effects were reported. Yeom and Heidrich[24] used a cross-sectional analysis of RCT data to report that communication difficulties with health professionals had significant direct, negative effects on QOL dimensions. Findings from a qualitative interview study highlighted the value to older patients with cancer of nurses having a person-centred manner, with the ability to show a genuine and empathic interest in the patients and to make a connection with good listening and communication skills.[23]

## DISCUSSION

This systematic review aimed to provide an overview of the evidence base on the effectiveness of workforce interventions for improving the outcomes for older people with cancer, as well as analysing key features of the workforce associated with those improvements. Findings reflect a range of ways in which the workforce has been adapted, expanded or trained to addressing older patients with cancer multiple and divergent needs. The findings present a novel synthesis of the type of interventions being developed globally to address the broad question of how the workforce can support the improvement of older people's cancer outcomes. The approaches and the patient problems they are addressing are varied, including integrating the input of geriatric specialists into cancer services, using APN roles to support patients, creating new roles to guide patients through the healthcare system and ensuring effective treatment, through to novel approaches using voluntary support, or trialling yoga or other exercise to improve older patients' QOL.

While the included studies begin to provide evidence about how the workforce can be used to make a tangible difference to physical and psychological outcomes of older patients with cancer, the diversity of interventions in the studies reviewed and the range of outcomes evaluated limit generalisations on effectiveness. Further, the quality of evidence is generally low. Experimental designs were not consistently used and, when they were, their implementation was often hampered by poorer than expected recruitment, or conclusions drawn about outcomes for older patients were drawn from a secondary analysis of a wider data set. In addition, as is common in the reporting of complex intervention evaluations, details of the intervention itself were often lacking.[45] There was inadequate reporting of the specific workforce contribution to the interventions and limited evidence to address the second objective of the review around the features of the cancer care workforce associated with better outcomes. In addition, while staff training was involved in half of the interventions reported, the details of how that training worked or could be improved were not detailed. Furthermore, although some innovative roles were set up, the rationale and detail of those roles were often poorly reported.

Despite these shortcomings, these findings do provide some promising insights into how the workforce may address the varied needs of older patients with cancer, although with a dearth of evidence at the earlier and later stages of the cancer journey. Evidence has suggested that not all older people with cancer need the same input, and indeed age-related changes occur at different rates in different individuals and are not reflected in chronological age.[7] Therefore, it is more productive to focus attention on those with complex problems.[46] The studies in this review appear to support the notion of targeted assistance to groups at particular risk of undertreatment. Review findings suggest that broader interventions aiming to improve survival outcomes are less successful, but studies did indicate the kind of support that could

be put in place after treatment to deal with the specific complications and problems that older people might face. One intervention which did improve survival used APN in home care support postsurgery.[37] Indeed, the role of APN in the future of older people's cancer care has been acknowledged elsewhere in the literature,[47–50] and this review indicates that this is a candidate role for exploration and further consideration.

The input of geriatric specialists who are able to assess and manage older patients and optimise patients for treatment was a significant feature of several studies reviewed and formal links and services are well established in some countries.[51–53] Findings from this review provide weak evidence of positive benefits from the input of geriatricians but it only included studies where the geriatrician's role was explicit in the intervention and where a comparison or control was featured. There are a number of other reviews reporting on specialist geriatric assessment and management for older patients with cancer, and these have been able to draw firmer conclusions about the benefits of CGA with older patients with cancer, although they all acknowledge the need for more definitive research.[54–56] Multidisciplinary approaches also emerged as a feature across the studies reviewed and the need to shape teams around the multiple needs of older people with cancer has been highlighted elsewhere, although evidence from this review is weak, again limited by the scale and quality of the research.[6 57–60]

Of further interest is the use of non-professionals in providing direct care services to older people with cancer, and roles such as these are relevant in the contexts of budgetary pressures and recruitment difficulties of key professional groups such as geriatricians and registered nurses.[17] The two studies reviewed suggested a positive impact on patient outcomes and align with a growing recognition of the non-clinical workforce (including carers and families) playing an essential role in older people's cancer care.[61–63] However, the low quality of the research again reduces confidence in these positive findings. A final point is that the studies identified for this review did not address the impact of staffing levels or skill mix on outcomes of older patients with cancer. In addition, few mechanisms to develop the current workforce to prepare for and be supported to deliver high-quality care to an ageing population were identified. In addition to the development and more definitive evaluation of new roles and practices, the future research agenda must address these important facets to ensure that, regardless of setting, all healthcare workers that older people with cancer encounter are prepared for and adequately supported in their role.[64 65]

This review alone is insufficient to enable conclusions to be drawn about the workforce factors which prove most beneficial to older people's outcomes; further high-quality RCTs are needed to assess the potential of possible interventions. Future research should build on the studies reviewed here to establish what workforce developments are needed to support this growing population throughout

the cancer journey. The most promising interventions for further study target assistance to individuals with complex needs who are at particular risk of undertreatment, and of problems arising from cancer treatment or its impact. Our review indicates that the impact of multiprofessional teams, including geriatric physicians and APN, on patient outcomes from survival to QOL, would be worthwhile to evaluate more definitively, as would the contribution of trained volunteers.

**Twitter** @JackieLearning

**Contributors** JB, GL, PG and TW were responsible for the systematic review design. GL was responsible for data collection. JB, GL and PG were responsible for data extraction and appraising studies. JB, GL, PG and TW were responsible for data analysis and interpretation. All authors contributed in drafting the manuscript. JB is responsible for the overall content as the corresponding author.

**Funding** This work was funded by Macmillan Cancer Support to support the work of the Expert Reference Group for the Older Person with Cancer. The views expressed are those of the authors and do not necessarily reflect those of Macmillan Cancer Support.

**Competing interests** None declared.

**Provenance and peer review** Not commissioned; externally peer reviewed.

**Data sharing statement** No additional data are available.

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
