## [Reviewer comments · BMJ Open]

ARTICLE DETAILS

TITLE (PROVISIONAL)	Workforce characteristics and interventions associated with high quality care and support to older people with cancer: a systematic review
AUTHORS	Bridges, Jackie; Lucas, Grace; Wiseman, Theresa; Griffiths, Peter

VERSION 1 - REVIEW

REVIEWER	Kerry Allen University of Birmingham, England, UK.
REVIEW RETURNED	10-Feb-2017

GENERAL COMMENTS	This paper makes a novel contribution to the field, is methodologically sound and well written. This reviewer suggest two key amendments: 1. Clarifying objectives at the beginning of the paper The objectives are communicated but they could be clearer. The research questions that guided the review design and synthesis are not clearly conveyed before moving into the methods section. The review seems to be asking two important questions: how are the workforce being used/supported to deliver older people's cancer care and is it working? Then, do we have the right workforce to deliver the best care to older people? The first objective around identifying and assessing workforce based interventions is clear. The secondary objective 'to analyse the features of the cancer care workforce associated with better outcomes for older people' feels more vague. What exactly are the 'features'? At this early stage it was difficult for this reviewer to think through the difference between the two objectives. There were potential overlaps e.g. staff training/knowledge – an intervention or a feature of the workforce? Either a much clearer explanation of the two objectives and the relationship between them, or the inclusion of distinct research questions to meet these objectives may help to convey the purpose of the review. At the beginning of the 'Discussion' section these objectives are better worded and you can see how the second objective links to the first, however this meaning may have got lost when drafting the abstract and background sections. 2. What sort of narrative synthesis? The nature and techniques of narrative synthesis within literature reviews have been interpreted in different ways. It would be helpful to include a methodological reference so that the reader can understand what you mean by narrative synthesis and what you did. This would increase the chances that your study could be repeated by others. This reviewer assumed your approach would be in line
---

	with Popay et al. 2006, endorsed by Cochrane, but useful to clarify. Other comments: -Excellent discussion section that introduces useful findings while acknowledging the insufficient nature of the review as a basis for generalisable conclusions. -Reviewing published literature there may be no important ethical issues or processes, but still might be worth clarifying this explicitly. Was all literature in the public domain?
--	---

REVIEWER	Schroder Sattar and Martine Puts Lawrence S. Bloomberg Faculty of Nursing, University of Toronto Toronto, Canada
REVIEW RETURNED	13-Feb-2017

GENERAL COMMENTS	Thank you for giving us the opportunity to review “Workforce characteristics and interventions associated with high quality care and support to older people with cancer: a systematic review”. We enjoyed reading the review but we have some questions/comments. In the strengths and limitations section it would be helpful to identify the type of review conducted. Introduction Could you please clarify the two research questions, I am not sure it are 2 or just one research question? Could you please clarify how you used 1 search of the literature to answer both questions? Methods Could you please clarify if an expert librarian was consulted for the search? How many persons reviewed each abstract and full text paper for eligibility? Inclusion criteria: how was the eligibility criterion “outcomes of interventions...which require significant workforce reconfiguration or the implementation of new roles” used when assessing papers? Would countries not be different in workforce configuration or the roles in use in providing cancer care to older adults and this thus different for different countries? And on page 5, there is also a section about eligible study types. Perhaps the inclusion criteria can be rewritten combining the bullets and the paragraph below to be clearer in the inclusion and exclusion criteria for this review. And could you please clarify why no limits on publication dates were used to ensure findings are still relevant to current cancer treatments and current practices? Could you please increase the font in Figure 1 the PRISMA flow chart? It is very hard to read. The reference to the GRADE system is incorrect, the first author of the GRADE system is Dr. Gordon Guyatt. Results First paragraph, please clarify if the 4,555 participants refers to patients or nurses?
---

	In your tables, please add effect sizes if available. Could you also prepare a table with all the studies and their characteristics? It is not possible to read the results section and particularly the quality assessment if it is not possible to understand the study design. Particularly country of study and study year, response rate of studies, methods used would be helpful to place the study into context. Could you please clarify how you decided to group the studies in the results section as you did vs organized by your two research questions? We are wondering about the grouping of studies as there is a real mix of study designs, study populations and studies conducted included pilot studies as well so we wonder how you decided to organize it this way? It is hard to take a message away from it. Discussion Please add the limitation that only studies published in English were eligible for inclusion. Please spell out all abbreviations under each table. Supplementary file 2: the data on the methods seems to be missing?
--	--

VERSION 1 – AUTHOR RESPONSE

Reviewer: 1

Reviewer Name: Kerry Allen

Institution and Country: University of Birmingham, England, UK.

Please state any competing interests or state 'None declared': None declared

Please leave your comments for the authors below

This paper makes a novel contribution to the field, is methodologically sound and well written.

This reviewer suggest two key amendments:

1. Clarifying objectives at the beginning of the paper

The objectives are communicated but they could be clearer. The research questions that guided the review design and synthesis are not clearly conveyed before moving into the methods section. The review seems to be asking two important questions: how are the workforce being used/supported to deliver older people's cancer care and is it working? Then, do we have the right workforce to deliver the best care to older people? The first objective around identifying and assessing workforce based interventions is clear. The secondary objective 'to analyse the features of the cancer care workforce associated with better outcomes for older people' feels more vague. What exactly are the 'features'? At this early stage it was difficult for this reviewer to think through the difference between the two objectives. There were potential overlaps e.g. staff training/knowledge – an intervention or a feature of the workforce?

Either a much clearer explanation of the two objectives and the relationship between them, or the inclusion of distinct research questions to meet these objectives may help to convey the purpose of the review. At the beginning of the 'Discussion' section these objectives are better worded and you can see how the second objective links to the first, however this meaning may have got lost when

drafting the abstract and background sections.

Response: we have clarified the objectives in the abstract and the background section.

2. What sort of narrative synthesis?

The nature and techniques of narrative synthesis within literature reviews have been interpreted in different ways. It would be helpful to include a methodological reference so that the reader can understand what you mean by narrative synthesis and what you did. This would increase the chances that your study could be repeated by others. This reviewer assumed your approach would be in line with Popay et al. 2006, endorsed by Cochrane, but useful to clarify.

Response: we have added a reference to indicate the methods used.

Other comments:

-Excellent discussion section that introduces useful findings while acknowledging the insufficient nature of the review as a basis for generalisable conclusions.

-Reviewing published literature there may be no important ethical issues or processes, but still might be worth clarifying this explicitly. Was all literature in the public domain?

Response: we have clarified that all literature was published journal papers.

Reviewer: 2

Reviewer Name: Schroder Sattar and Martine Puts

Institution and Country: Lawrence S. Bloomberg Faculty of Nursing, University of Toronto, Toronto, Canada

Please state any competing interests or state 'None declared': None declared

Please leave your comments for the authors below

Thank you for giving us the opportunity to review "Workforce characteristics and interventions associated with high quality care and support to older people with cancer: a systematic review". We enjoyed reading the review but we have some questions/comments.

In the strengths and limitations section it would be helpful to identify the type of review conducted.

Response: this has been added.

Introduction

Could you please clarify the two research questions, I am not sure it are 2 or just one research question? Could you please clarify how you used 1 search of the literature to answer both questions?

Response: the research questions have been clarified in the abstract and the background section.

Both questions shared the same key search terms and so it was possible to use the one search for this combined purpose to yield material that would answer one or both of the questions.

Methods

Could you please clarify if an expert librarian was consulted for the search? How many persons reviewed each abstract and full text paper for eligibility?

Response: we did not consult an expert librarian for the search as team members have a significant track record in designing and executing complex searches in relation to these topic areas.

Explanations of how many people were involved in review for eligibility can be found on p.7. One author reviewed all abstracts and about 5% of all abstracts were reviewed by one of the remaining three authors. All full text papers were reviewed by GL plus one of the other authors.

Inclusion criteria: how was the eligibility criterion "outcomes of interventions...which require significant workforce reconfiguration or the implementation of new roles" used when assessing papers? Would countries not be different in workforce configuration or the roles in use in providing cancer care to

older adults and this thus different for different countries?

And on page 5, there is also a section about eligible study types. Perhaps the inclusion criteria can be rewritten combining the bullets and the paragraph below to be clearer in the inclusion and exclusion criteria for this review.

Response: we have revised the explanation for inclusion/exclusion criteria in line with your suggestions (pp.6-7).

And could you please clarify why no limits on publication dates were used to ensure findings are still relevant to current cancer treatments and current practices?

Response: Rationale have been added on p.5.

Could you please increase the font in Figure 1 the PRISMA flow chart? It is very hard to read.

Response: we have increased the size of the chart so it is now easier to read.

The reference to the GRADE system is incorrect, the first author of the GRADE system is Dr. Gordon Guyatt.

Response: we have now corrected this.

Results

First paragraph, please clarify if the 4,555 participants refers to patients or nurses?

We have clarified this in Results section (p.9) and the abstract (p.2).

In your tables, please add effect sizes if available.

Response: We have supplied effect sizes in original units where this information is supplied and suggest that this is more meaningful in the context of individual studies with their own selection of outcome measures than providing standardised effect sizes.

Could you also prepare a table with all the studies and their characteristics? It is not possible to read the results section and particularly the quality assessment if it is not possible to understand the study design. Particularly country of study and study year, response rate of studies, methods used would be helpful to place the study into context.

Response: we have completely revised the tables in the results section and all of the core requested information is now presented in one table (pp.17-21).

Could you please clarify how you decided to group the studies in the results section as you did vs organized by your two research questions?

We are wondering about the grouping of studies as there is a real mix of study designs, study populations and studies conducted included pilot studies as well so we wonder how you decided to organize it this way? It is hard to take a message away from it.

Response: we have added an explanation about why we grouped the studies in this way (p.10) and also refined the Discussion to more clearly draw on this classification (pp. 14 and 16).

Discussion

Please add the limitation that only studies published in English were eligible for inclusion.

Response: we have added this to strengths and limitations section (p.3)

Please spell out all abbreviations under each table.

Response: we have added this (p.21).

Supplementary file 2: the data on the methods seems to be missing?

Response: we have added this (pp.31-32).

VERSION 2 – REVIEW

REVIEWER	Kerry Allen University of Birmingham, UK
REVIEW RETURNED	04-Apr-2017

GENERAL COMMENTS	All comments and suggestions responded to in full. Reads as a much improved version with interest to wide audience.
---

REVIEWER	Schroder Sattar and Martine Puts Lawrence S. Bloomberg faculty of Nursing, University of Toronto, Toronto, Canada
REVIEW RETURNED	05-Apr-2017

GENERAL COMMENTS	Figure 1 is missing in the revision.
--------------------------------------

VERSION 2 – AUTHOR RESPONSE

The remaining issue identified by reviewers is that Figure 1 is missing from the manuscript. This was removed from the main document at the request of the journal's team and has been submitted as a separate file.